# 3D Genome Organization: Causes and Consequences for DNA Damage and Repair

**DOI:** 10.3390/genes13010007

**Published:** 2021-12-21

**Authors:** Ànnia Carré-Simon, Emmanuelle Fabre

**Affiliations:** Université de Paris, CNRS UMR7212, INSERM U944, Génomes Biologie Cellulaire et Thérapeutiques, F-75010 Paris, France; annia.carre-simon@inserm.fr

**Keywords:** chromosome organization, double strand break, chromatin dynamics

## Abstract

The inability to repair damaged DNA severely compromises the integrity of any organism. In eukaryotes, the DNA damage response (DDR) operates within chromatin, a tightly organized DNA–histone complex in a non-random manner within the nucleus. Chromatin thus orchestrates various cellular processes, including repair. Here, we examine the chromatin landscape before, during, and after the DNA damage, focusing on double strand breaks (DSBs). We study how chromatin is modified during the repair process, not only around the damaged region (in *cis*), but also genome-wide (in *trans*). Recent evidence has highlighted a complex landscape in which different chromatin parameters (stiffness, compaction, loops) are transiently modified, defining “codes” for each specific stage of the DDR. We illustrate a novel aspect of DDR where chromatin modifications contribute to the movement of DSB-damaged chromatin, as well as undamaged chromatin, ensuring the mobilization of DSBs, their clustering, and their repair processes.

## 1. Introduction

The impact of genome organization on DNA transactions began to be appreciated when it became clear that genomes were not totally randomly organized, but followed organizing principles [1,2,3,4]. The spatial arrangement of eukaryotic genomes is thus based on two distinct scales of organization. At the scale of the chromosome, the attachment of different loci to the nuclear envelope (NE) and molecular affinities between different genomic regions shape chromosome organization. At the scale of the chromatin, in addition to nucleosomal landscape, chromatin loops result from the extrusion activity of structural maintenance of chromosome (SMC) complexes (consisting of cohesins, condensins, and Smc5/6) [5,6,7]. However, the mechanistic links between structure and function are still in their infancy, especially when DNA damages occur. Such damages can be due to chemical, physical, or biological factors including ultraviolet light or X-radiation, the incorporation of incorrect bases during DNA replication, or the presence of free radicals because of metabolic pathways. DNA lesions are of very diverse natures: altered or lost bases, intra- or inter-strand links, thymine dimers, or single or double strand breaks (DSBs). DSBs are the most deleterious and represent a challenge to cell survival. Many sources of DSBs are accidental. Whether exogenous or associated with cellular metabolism, these DSBs can cause translocations, deletions, insertions, aneuploidies, or even cell death if they are poorly or not repaired [8,9,10]. However, some DSBs are necessary for normal cell development under controlled physiological conditions. This is the case for meiotic DSBs, those involved in VDJ recombination during the immunological repertoire arrangement or during the change of sexual type in hemiascomycetous yeasts. While the fine coordination of DSB detection, signaling, and repair is better understood, the impact of chromatin architecture and chromosome spatial organization is just beginning. A large field of research now concerns the interplay between genome organization and its stability. With this review, we interrogate how genome organization might influence susceptibility to damage and modes of repair and conversely how damages affect and cause changes in chromatin organization. We begin by outlining how organization of chromosomes is based on (i) the SMC complexes that have a preponderant role both at the chromatin level and more globally on the folding of chromosomes and (ii) anchoring at the nuclear periphery.

## 2. Genome Organization without Damage: The Interplay between the Chromatin Scales

### 2.1. Chromosome Territories and Rabl-like Chromosome Configurations

At a large scale, only two types of chromosome spatial organizations are found during the interphase in eukaryotes: chromosome territories (CTs) and the Rabl-like chromosome organization. In the first mode, chromosomes are organized in territories confined in subnuclear areas with poor intermingling between the territories except at their boundaries [11,12]. The so-called Rabl-like configuration is characterized by centromeres clustering and telomeres anchoring to the NE (Figure 1). Rabl has been first described in budding yeast [13,14,15], fission yeast [16], Drosophila [17,18], and plants [19]. A recent study reports that these two types appeared several times independently during evolution in phylogenetically distant species [20].

Central to chromatin organization are the structural maintenance of chromosome (SMC) complexes. SMCs are evolutionary conserved families that share a similar ring-shaped structure with two SMC coiled–coil proteins, forming the core structure and a kleisin that bridges the two SMC subunits’ ATPase heads (Figure 1a, left). Cohesins are well known to encircle sister chromatids during meiosis and mitosis, avoiding sister chromatid separation when microtubules pull centromeres at metaphase/mitotic transition. The budding yeast cohesin complex is composed of the two Smc1 and Smc3 subunits and the kleisin Scc1 (also known as Rad21 in mammals) that links both Smc1 and Smc3 together. There are several additional regulatory subunits such as Scc4, Pds5, Wpl1, and Eco1 [21] (Figure 1a, right). Condensins exist in two types in vertebrates: condensin I and condensin II [22]. The two complexes present the same SMC proteins (Smc2 and Smc4), while the non-SMC proteins differ: condensin I includes CAP-H-D2 and -G whereas condensin II contains CAP-H2-D3 and -G2 [22] (Figure 1a, middle). In yeasts such as *Saccharomyces cerevisiae* and *Schizosaccharomyces pombe*, only the condensin I is found [23].

Interestingly, it was observed that the presence or absence of condensin II is intimately linked to the formation of CTs or the Rabl configuration, respectively [20,24]. Initially, it was discovered that the absence of CAP-H2 in Drosophila ovary nurse polyploid cells prevented the formation of CTs normally observed at the endoreplicative stage 10 of these cells [24]. The link between condensin II and CTs is now documented beyond the Drosophila case [20]. A Hi-C study of genomic organization in 24 species covering the eukaryotic kingdom revealed a correlation between Rabl configuration and absence of CAP-H2. Moreover, it was shown that in human cells, organization of CTs can convert into a Rabl configuration upon deletion of CAP-H2 [20] (Figure 1b). These results highlight the relationship between a family of the SMC complexes and chromosome organization, further indicating that, throughout evolution in eukaryotes, only these two modes of 3D chromosome organization appear to exist, depending on condensin II. How can we explain the evolutionary transitions between Rabl organization and CTs? It is possible that condensin II was lost several times during evolution, allowing a Rabl configuration repeatedly. The maintenance of the Rabl configuration in some organisms may in turn suggest a possible selective advantage over CTs. Alternatively, condensin II differs from condensin I by only three subunits [22]. Condensin II complex formation could be facilitated in some organisms, giving rise to CTs. To understand the nature of the interplay between these two types of chromosome organization, the expression of condensin II subunits in organisms with a Rabl configuration, such as yeast, could prove useful.

### 2.2. The Role of Cohesins in Chromosome Compartment Organization

Unexpectedly, the cohesin complex also appears to participate in the spatial organization of chromosomes (Figure 1b). A characteristic pattern of chromosome organization in a number of mammals results from enriched or depleted interactions in *trans* (between chromosomes) and in *cis* (along the fiber). Named A and B, depending on the level of chromatin wrapping [25,26], these compartments have the particularity of being highly correlated to transcriptional activity and to certain histone marks. Compartment A corresponds predominantly to euchromatin, enriched in actively transcribed genes, while compartment B represents heterochromatin where genes are in a more repressive state [25,26,27]. Effective depletion of cohesin (∆Scc2 or ∆Scc4) or Wpl1, which increases the residence time of cohesin on chromatin, results in strengthened or weakened compartments respectively, as detected by Hi-C [28,29,30]. These results indicate that compartments can form by a cohesin-independent mechanism, but cohesin activity is able to counteract domain-specific attractions. Cohesins would have a role in creating long-range interactions between euchromatin or heterochromatin regions. How do compartmentalization and cohesin activity cooperate? Simulations based on polymer physics and attractive forces between euchromatin and heterochromatin [31] suggest that attraction between compartments enables phase separation, while cohesin activity actively participates in compartment mixing. Genome organization into compartments would therefore be a trade-off between phase separation and cohesin activity, i.e., loops extrusion.

Compartments are not evidenced as such in eukaryotes presenting a Rabl configuration, such as budding yeast *S. cerevisiae*. Yet, cohesin and condensin do play a role in 3D chromosome folding in remarkably different ways than their canonical roles in mammals. During mitosis, cohesins, but not condensins, compact chromosome arms by creating short-range loops. In contrast, condensins are necessary for the compaction of the rDNA locus and organize centromeres by isolating them from the rest of the chromosomes arms, as demonstrated by Hi-C and polymer modeling [32,33].

**Figure 1 genes-13-00007-f001:**
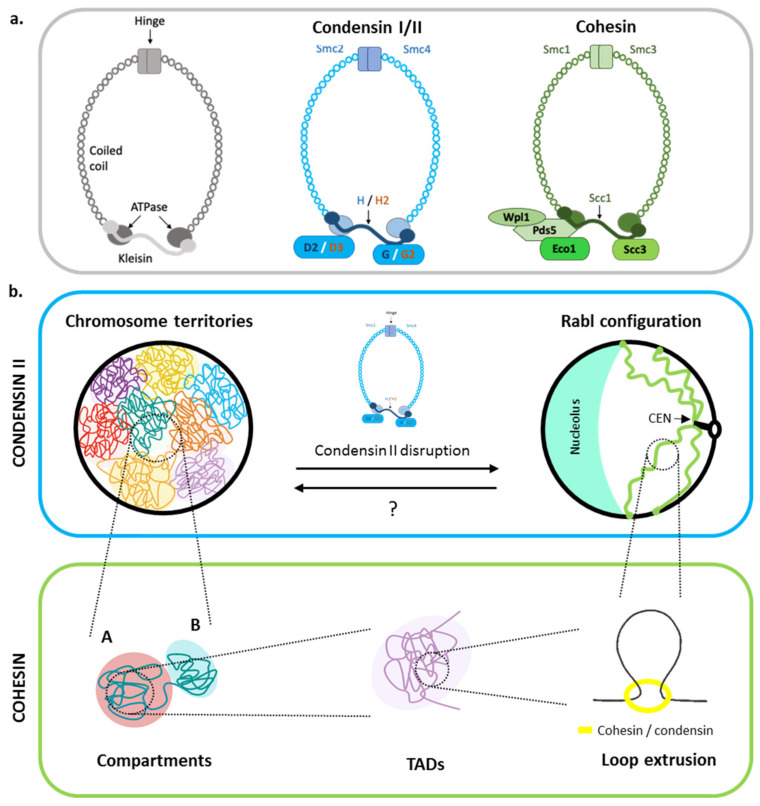
Chromatin organization and the role of SMC complexes. SMC complexes organize the chromatin architecture at different levels. (**a**) Left: Overview of the SMC complex in *S. cerevisiae*. The SMC subunits have a coiled–coil domain that forms a dimmer (hinge) and the ATPase held together by a kleisin. Middle: Condensins I (blue subunits) and II (orange subunits) are composed of Smc2 and Smc4 held together by kleisin H in condensin I, and H2 in condensin II. In addition, the condensins have a pair of HEAT subunits, named D2 and G in condensin I and D3 and G2 in condensin II [22]. Right: The cohesins (green) are formed by Smc1 and Smc3 linked by Scc1. It has regulatory subunits such as Scc3, Wpl1, Eco1, and Pds5 [21]. (**b**) Condensin II organizes chromosomes on a large scale. Its presence allows for the organization of chromatin into chromosomal territories. The absence of this complex results in the organization of chromosomes in a Rabl configuration [20]. On a smaller scale, cohesins are the main actors. This complex controls the organization of chromatin into two compartments, named A and B, in mammals [28,29,30]. There is no evidence of the presence of these compartments in Rabl-like organisms. In addition, cohesins allow the formation of TADs by extruding the DNA and creating loops [27,34,35]. The organization of chromatin in TADs in species with the Rabl-like configuration is still under debate.

### 2.3. Loop Extrusion

The extreme improvement in resolution of 3C techniques, and its 5C and Hi-C derivatives, enabling even nucleosome-scale resolution (Micro-C), offers a detailed description of subchromosomal organization [36]. In mammals, these intra-chromosomal domains have been named topological associated domains (TADs). TADs are characterized by more enriched contacts in close proximity than with other parts of the genome, as demonstrated in human cells [34], mice [35], and Drosophila [27]. These domains are functionally important because they can bring distant elements of the genome together, such as enhancers and their targets, thereby influencing gene expression [37,38]. The presence of TADs in other species such as yeast is still questioned, although two types of domains, one 250 kb in size with different replication timing [39] and another 2–10 kb in size, including 1-5 genes [40,41], have been found.

The interactions favored within a TAD and limited with neighboring TADs indicate the presence of boundary regions. In mammals, these barriers are enriched in architectural proteins such as CCCTC binding factor (CTCF) and cohesins, as well as highly transcribed genes and tDNA [34,35]. CTCF has only been described in vertebrates and Drosophila [42]; the absence of CTCF in budding yeast may suggest that boundaries cannot be established. Nevertheless, domain boundaries have been observed by Hi-C even in budding yeast [41]. They occur at highly transcribed genes where high levels of ATP-dependent chromatin remodeling complex RSC and cohesins are found [41]. Importantly, microscopy observations coupled with Hi-C observations in single cells show that TADs are not stable structures but reflect highly variable contacts enriched in a small number (~10%) of cells at a given time [43,44].

One model to explain TAD formation is loop extrusion, a process in which motor protein complexes bind to DNA and “pull” chromatin, gradually creating loops. The loop creates a contact between two distant sites along the fiber and appears as a “dot” on Hi-C maps [45]. Loop extrusion involves two of the SMC complexes: cohesins and condensins. Cohesins organize chromosomes by loop extrusion in mammals [46,47] and yeast [48,49] (Figure 1b). The length of the loop depends on several factors, such as the amount of cohesins and their residence time on chromatin [48]. In yeast, cohesins have been shown to be recruited at centromeres and in regions where transcription of two genes is convergent [50]. Loop size would therefore regulate interactions along a single chromosome.

The enrichment of cohesins on chromatin depends on two main players, Wpl1 and Pds5, which operate in different ways. While Wpl1 acts as a negative regulator by removing cohesins from chromatin, Pds5 binding to cohesins keeps them bound to it. Depletion of either Wpl1 or Pds5 paradoxically reduces the number of cohesins genome-wide, increasing loop length in both contexts [28,30,48,49,51]. Upon depletion of Wpl1, one would have expected an increase in chromatin-bound cohesins throughout the genome and, therefore, a decrease in loop size, but Hi-C shows the opposite [48]. The increase in loop size could be explained if Wpl1 could remove nonspecific binding of cohesins to chromatin [48]. Upon depletion of Wpl1 in mammals, an accumulation of cohesins has also been observed but in elongated chromatin structures called “vermicelli” that would form the base of the loops [28,30,52]. It is therefore tempting to think that a similar regulation of cohesins exists in mammals and yeast where, upon deletion of cohesin regulatory proteins, the enlarged size of loops would be explained by an increase in cohesins at specific sites and a decrease at less specific sites. This would allow in *cis* longer-range interactions at the genome scale.

In summary, it is clear that SMC complexes are key factors for chromatin organization at different levels. This function of SMCs in shaping chromosomes is remarkably conserved and extends to all kingdoms of life, including bacteria and archaea [53,54,55]. Although our knowledge of 3D genome architecture is rapidly increasing, we still need further experiments, such as resolutive Hi-C as well as single molecule tracking in mutants of SMC complexes, to understand the enrichment of SMCs along chromosomes, the dynamic formation of loops, and how this level of chromatin organization can regulate large-scale genome organization. The respective impacts of these different layers of organization on genome integrity are discussed in the following sections.

### 2.4. Nuclear Envelope Tethering: A Prominent Role in Genome Organization

Tethering to nuclear structures plays an additional important layer in genome configuration. The nuclear periphery is thus the most prominent landmark where chromatin is often in the form of heterochromatin. In metazoans, the nuclear lamina, a fibrillar network near the inner nuclear membrane, is a key factor for the organization of peripheral chromatin. Genomic regions that interact with the nuclear lamina, so-called lamina-associated domains (LADs), were identified using the DamID technology [56]. LADs are found conserved among mammals and rely on association with lamin B and A/C. The association with lamin is critical for the peripheral positioning of heterochromatin, since in the absence of lamin receptors, peripheral tethering is lost and nuclei appear inverted [57]. In budding yeast, with Rabl-like chromosome organization, centromeres, and telomeres are the genomic domains that lie close to the nuclear periphery. The mechanisms by which attachment to the periphery occurs are not well known, although protein amount and protein–protein attraction are critical. One example in budding yeast is the silent information regulatory proteins or SIRs (Sir1, Sir2, Sir3, and Sir4) that accumulate at chromosomal ends [58]. Disruption of Sir4 binding to chromatin promotes detachment to the nuclear envelope [59]. Similarly, modeling of protein attraction can modulate attachment to the nuclear periphery [60].

## 3. Are Some Genomic Regions More Fragile Than Others?

The architecture of the genome is thus sophisticated and finely regulated. Given that the genome is subject to multiple insults, which damage DNA, the question arises as to whether genome architecture or sequence may contribute to differential susceptibility to DSBs, in other words, whether the genome is uniformly sensitive to damage. Indeed, certain genomic regions appear to be more prone to endogenous DSBs than others, such as heterochromatin, ribosomal DNA, or chromatin loops [61,62,63,64]. The fragility of these regions has been attributed to several features that can transmit replication stress such as repetitive sequence content, premature mitosis, or late replication timing or paucity of replication origins [65]. In addition to replication, transcription is itself considered to be a source of DSBs. In fact, it has been shown that DSB frequency was elevated in particular near transcriptionally active genes, with a significant enrichment around the transcription start site (TSS) [66,67]. Although the precise mechanism underlying DSB-induced transcription activation remains elusive, the transient strand separation during transcription is thought to render the non-transcribed strand particularly vulnerable. New evidence has attributed these transcription-induced DSBs near activated genes to the action of type II topoisomerase (TOPII) [61,68]. This type II topoisomerase, by introducing DSBs, can relax DNA torsional stress due to the transcriptional machinery [61,68,69]. Furthermore, persistent DSB introduced by TOPB2B and left unresolved can lead to oncogenic translocations [68]. For instance, loop boundary regions enriched at the highly active mixed-lineage leukemia (MLL) gene are enriched in TOP2B. Therefore, induction of these DSBs results in recurrent translocations with other similarly damaged genes, ultimately leading to acute myeloid leukemias [68].

Besides 3D chromatin organization as a factor in DSB formation, nucleotide sequence can also be a cause of genome fragility. Expansion of repeated sequences such as trinucleotide repeats (TNRs) are often associated with human diseases. TNRs are capable of creating secondary structures such as loops and hairpins due to the pairing of complementary single-stranded DNA. The formation of these secondary structures is prone to the collapse of replication forks resulting in DSBs [70,71,72]. The level of fragility of TNR regions is correlated with the number of triplets: the more trinucleotide repeats, the higher the fragility of the region [70,72].

These examples highlight how genomes, through their chromosomal folding or DNA content, can be a source of genomic fragility. Henceforth, future experiments may focus on the predictive power of chromosome conformation in determining the hotspots of DSBs.

## 4. What Is Happening to the Chromatin When a DSB Occurs?

Studies to date indicate that the non-random organization of chromatin in the nucleus may have a role to play in genome fragility. The next question we ask is how chromatin architecture changes during DSB; we use experimental data obtained in yeast and mammals to illustrate our point.

### 4.1. Chromatin Architecture Changes: Role of Cohesins

DSBs induce changes in chromatin in the vicinity of the DSB and genome-wide. One of the first and rapid changes in chromatin after a DSB is the phosphorylation of H2A or its mammalian variant H2AX (γH2A(X)), which extends from the site of DSB and occupies approximately one-third of the damaged chromosome arm [73,74,75,76]. Cohesins have been shown to follow this pattern in a γH2A(X)-dependent manner [77,78]. However, the role of cohesins in DSBs and their subsequent repair has remained unclear until recently. It was found in mammalian cells that cohesins were enriched at DSBs generated by the endonuclease *AsiSI* in a manner mediated by NIPBL (the cohesin loader) and ATM [79]. Interestingly, DSBs act as a barrier for loop extrusion, as evidenced by Hi-C detection of “stripes”. In addition, 4C-seq, which visualizes contacts from one viewpoint (here, the DSB), indicates that loop extrusion enhances contacts between DSBs and neighboring sequences (Figure 2a). ChIP-seq against yH2A(X) in synchronized cells also indicates asymmetric spreading of yH2A(X) from the DSB in a cohesin and loop size-dependent manner. It has been proposed that the DSB caused a one-side loop extrusion by enriched cohesins near the break. Loop extrusion, along with phosphorylation of H2A(X) by ATM, propagates until it reaches the boundary of a TAD, confining the damage within a TAD, thus providing an efficient means of signaling damage and creating a specific compartment conducive to repair [79].

Using the Hi-C technique, the impact of a DSB on chromatin folding was also addressed in budding yeast [51]. Hi-C maps show that after 4 h of HO endonuclease induction, chromatin reorganizes. Together with a local decrease in the vicinity of the DSB, contacts between ≈25 kb regions flanking the DSB are detected. This structure, formed by the contacts between flanking double-stranded DNA, depends on MRX resection but arises independently of the cohesin, Mec1 (ATR), Rad52, or Rad51. Furthermore these contacts may block cohesin progression, as visualized by a border pattern [51]. Globally, the damaged chromosome shows a looped organization typical of structures found in G2/M. This last profile implies an isolation of the chromosome, therefore a decrease in *trans* contacts and a chromosome topology favorable to the repair of the break in *cis* (Figure 2a).

Thus, cohesins play an important role in genome modification after DSB, both globally and locally. In both yeast and mammals, cohesins create loops and limit genome-wide interactions upon damage, which reduces ectopic repair by (1) looping chromosomes and promoting *cis* interactions in yeast and (2) limiting homology search to a single TAD in mammals. Further studies are needed to understand in depth the role of chromatin organization in DSB repair. Thus far, it is known that cohesins contribute to its repair, but some questions remain unresolved, such as the involvement of other SMC complexes in DSBs. For example, SMC5/6, one of the lesser-known SMC complexes, is recruited to both sites of DSB, as demonstrated by chromatin immunoprecipitation (ChIP) [80,81,82]. Depletion of SMC5/6 results in cell hypersensitivity to a broad spectrum of DNA-damaging agents, indicating its role in damage [83,84]. An interesting hypothesis is that Smc5/6 cooperates with cohesins in chromatin organization in the presence of damage.

**Figure 2 genes-13-00007-f002:**
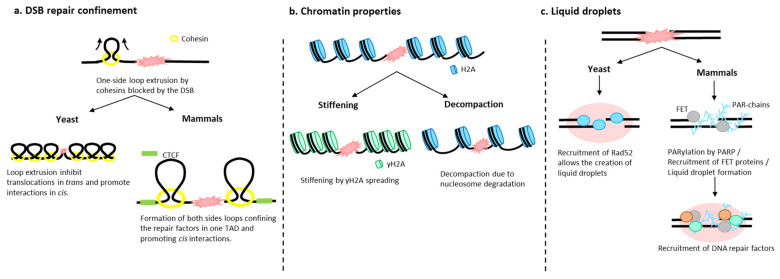
Factors that influence chromatin organization upon damage. (**a**) In yeast and mammals, upon damage, cohesins extrude DNA, creating a loop on one side of the damaged locus, as the DSB blocks cohesin progression. In yeast, cohesin loops cause chromosome individuation and limit the ability of DSB ends to explore their environment by decreasing *trans* contacts and favoring *cis* break repair [51]. In mammals, loop extrusion stops at DSB boundaries, confining repair proteins within a TAD [79]. (**b**) Chromatin undergoes intrinsic changes after a DSB such as stiffening by H2A(X) (γH2A(X)) phosphorylation, decompaction due to histone degradation, or condensation [85,86,87,88,89,90]. However, the intrinsic properties of chromatin after damage are not yet clear. (**c**) Upon damage, PARP-mediated PARylation is one of the first modifications observed in mammalian cells that allows the recruitment of FET proteins. The intrinsic highly structurally disordered domains of these proteins promote the formation of liquid droplets [91]. In yeast, the recruitment of repair proteins such as Rad52 are at the origin of liquid droplets, a likely means of isolating the DSB from the nuclear environment and concentrating some repair proteins [92,93].

### 4.2. Chromatin Fiber Changes upon DSB

In addition to local chromatin changes at DSBs and their connections to SMCs, intrinsic changes in chromatin fiber properties are also reported. It has been proposed that negative charges on histone H2A, phosphorylated on its amino-terminal tail after damage, may cause an increase in chromatin stiffness due to the repulsion between the nucleosomes [85,86] (Figure 2b). The fact that a phosphomimetic mutant of H2A causes a chromatin behavior compatible with stiffening supports this interpretation [94]. Similarly, Rad51 binding to the single strand formed after 5′ to 3′ resection from the DSB is at the origin of a nucleoprotein filament compatible with chromatin stiffening [87]. The increase in stiffness is manifested by an increase in mobility at different regimes of both the damaged locus and the rest of the genome. It has been proposed that during DSB, the stiffness of the Rad51 nucleoprotein filament may help donor search by acting as a “needle in a ball of yarn” in the crowded environment of the nucleus [87]. Other types of chromatin modifications have also been reported, such as condensation, observed after radiation damage [88]. This condensation is necessary and sufficient to activate and recruit DDR factors [88]. Finally, it has been proposed that during a DSB, chromatin decompaction associated with histone degradation after Zeocin multiple damages is a source of increased chromatin mobility [89,90] (Figure 2b). These chromatin modifications are not mutually exclusive, as compaction occurs just after a DSB to stabilize the damaged ends and concentrate DDR factors, while decompaction and stiffening may occur afterwards to allow DSB repair. Nevertheless, these different interpretations in fiber property changes are often based on polymer models, whose parameters cannot be fully verified in vivo. The ability to observe the chromatin fiber with super-resolution tools will undoubtedly be the means to determine the actual changes in chromatin in response to DNA damage.

### 4.3. Role of Liquid–Liquid Phase in DSB Foci

Recruitment of checkpoints and repair proteins can lead to the formation of microscopically detectable foci, as has been observed repeatedly in different eukaryotes. One function of these foci could be to locally increase the concentration of repair-promoting proteins, as it has been proposed in yeast [95]. Damaged DNA aggregation in a localized focus could also promote spatial proximity favorable to HR repair, as proposed in mammals [96]. Recent high-resolution microscopy studies have demonstrated that in mammals and yeast, these foci form membrane-free subcompartments around the DSB, also known as liquid droplets or liquid phase separation. Liquid droplets are enriched with RNA and intrinsically disordered proteins that contain low complexity domains that tend to phase separate [97]. In mammals, liquid phase compartments are seeded in a poly-ADP-ribose polymerase (PARP)-dependent manner. The negative charges due to parylation recruit FET proteins, which present high intrinsic structural disorder domains [91]. By time-lapse imaging, it is found that FET proteins undergo fusion and fission events, as well as the formation of spherical intracellular droplets interpreted as a phase separation. In yeast, it was observed by single particle tracking (SPT) and photo-activated localization microscopy (PALM) that some repair proteins, such as Rad52, present liquid droplet capacity since it exhibits a high level of intrinsic disorder [92,93] (Figure 2c).

An attractive hypothesis is that liquid droplets mediate the earliest response to DNA damage, organizing the soluble nuclear space and recruiting specific repair proteins. Once the damage is repaired, they disintegrate. However, questions remain, such as the mechanism by which the compartments are maintained or how some proteins are enriched while others are excluded.

## 5. How Does Genome Organization Affect Repair?

### 5.1. In the Choice of the Repair Pathway

Two main pathways have evolved to repair a DSB: non-homologous end joining (NHEJ) and homologous recombination (HR). While classical NHEJ repair is accomplished by ligating the two broken DNA ends, HR uses a homologous template elsewhere in the genome to repair the break, typically the sister chromatid during the G2 phase of the cell cycle. Many studies have focused on the involvement of the level of chromatin folding, euchromatin, or heterochromatin in the choice of repair pathway [98,99]. The revolution in CrispR/Cas9 genome editing certainly contributes to a comprehensive view of repair mode as a function of chromatin folding [100]. It was understood early on that active transcriptional units would promote HR, where the histone modification H3K36me3 is located [96,101]. Paradoxically, transcriptionally inactive regions of the genome enriched in the di- or trimethylated heterochromatin mark H3K9 (H3K9me2/3) have also been implicated in promoting HR [102]. Nevertheless, shifting the heterochromatin to euchromatin states of a single locus in mouse and Drosophila does not seem to imply a major change in the balance between NHEJ and HR, underscoring the need for more comprehensive approaches [63]. Recently, in order to exonerate from sequence effects and consider only chromatin context effects, Schep et al. (2021) used a reporter integrated, in human cell lines, in more than 1000 different random positions in the genome. After induction of DSB in the reporter by Cas9, multiplexed sequencing of the repair “scars” identifies the mode of repair [103]. This study shows that euchromatin is predominantly biased towards NHEJ repair, while alternative end joining pathways are found in heterochromatin. However, this system does not detect faithful repairs such as HR and exact NHEJ, and it is not yet known as to whether Cas9-induced DSBs behave like the DSBs studied thus far.

In addition to the folding effect, the spatial positioning of the DSB within the nucleus may also play a role in repair choice. The first study in mammalian cells using a locus artificially anchored to the nuclear lamina observed that a DSB induced therein was preferentially repaired by NHEJ, contrary to a DSB tethered to the nuclear pore complexes prone for repair by HR [104]. In yeast, two factors influence endonuclease-induced DSB repair: the proximity of the DSB to the nuclear envelope [15,105,106] and the 3D distance between DSB and homologous sequences [107,108]. Thus, the Rabl configuration dictates the efficiency of DSB repair, with DSBs close to their homologous sequence in nuclear space repairing more efficiently than those far from it [107]. Given the Rabl configuration of yeast chromosomes, a model based on polymer physics is able to predict repair efficiency by HR. However, this is without taking into account chromatin mobility, whose importance has recently been understood.

### 5.2. In Chromatin Mobility

Chromatin mobility is an important parameter to take into account in both repair pathways. During NHEJ, the most attractive hypothesis is that chromatin mobility increases the probability of encounter between broken DNA strands [109]. It has even been proposed that this mobility could prevent the formation of aberrant translocations [94,110]. This last point, however, is in apparent contradiction with an observed higher translocation frequency when chromatin is mobile [111]. The low frequency and long time taken for translocation formation, however, are in favor of a more efficient NHEJ in the short term. During homologous repair, chromatin mobility would be a means to facilitate the access of repair proteins to the damaged locus and make the encounter with the homologous sequence more efficient. The first evidence of chromatin mobilization was, paradoxically, obtained in fixed cells [112]. After α particle radiation damage, H2AX labeling, initially detectable along the radiation array, concentrates over time in distinct foci, indicating a mobility of the phospho H2AX chromatin. The absence of foci formation in ATM-deficient patients already pointed to the prominent role of this factor in the probable chromatin mobility. Since then, numerous studies in yeast have confirmed the involvement of checkpoint proteins, chromatin structure [85,87,89], molecular motors such as actin [113], and that of nuclear anchors [114,115].

Little is known about the impact of chromosome territories or Rabl chromosome organization on mobility. Pioneering work in yeast in 2001 already indicated that nuclear constraints resulted in reduced mean squared displacements of loci located in these constrained spaces [116]. These results have since been widely confirmed in yeast and also in mammalian models, but the precise consequences on DSB repair of these various constraints are not known. A striking example however of the impact of genome organization in mobilizing chromatin when it is damaged, is when damage occurs in repeated sequences. We have seen above that a DSB in the yeast rDNA repeats is relocated outside the nucleolus [64]. Similarly, induction of single or multiple DSBs in Drosophila and mammalian cell pericentromeric repeats involves mobilization of DSBs out of heterochromatin [63,117,118]. This genomic displacement is a prerequisite for HR repair, presumably to avoid recombination between repeats that could be a source of genomic instability. These heterochromatin DSBs generally move to the nuclear periphery in an actin-, myosin-, and Smc5/6 complex-dependent manner [119]. Furthermore the increased mobility observed in cohesin mutants, upon degradation of the Scc1 subunit for example, implies that loops limit the nuclear space explored by DSB [90,120]. This observation is consistent with the increased contacts in *cis*, around the damage, due to cohesins [51,79]. The creation of loops by cohesins thus might reduce chromatin mobility for faithful repair near the break and implies that other mechanisms might exist when search for homology far from the DSB is required.

## 6. Conclusions

Deciphering the relationship between chromosome organization and genome activity is essential for understanding genomic processes such as gene expression, maintenance of genome stability, or deregulation in diseases (Figure 3).

SMC complexes appear to be key players in genome architecture at different layers, at the chromosome level, by creating chromosome territories or Rabl-like configurations, or at the chromatin level, by giving rise to TADs through loop formation. Although it is not yet known as to whether other mechanisms exist in the folding of chromosomes and the organization of the genome, the involvement of this organization in the formation of DSBs, but also their repair, is beginning to emerge. The involvement of SMC complexes in DSB repair shows in this respect how chromatin structure is essential to safeguard genome integrity. Thus, it is noteworthy that mutations in SMC complexes are implicated in human diseases, e.g., mutations in the cohesin complex subunits can lead to Cornelia syndrome or mutations in the Nse3 subunit of Smc5/6 causes the lung disease chromosome breakage immunodeficiency syndrome (LICS) [121,122]. Our understanding of the links between genome organization and DSB repair will undoubtedly benefit from single molecule imaging and Hi-C techniques, taking into account the dynamic modifications of damaged chromatin, with this new parameter having a role to play in the response to damage. Understanding whether and how organization is involved in the balance between faithful maintenance of genomic information and its evolution is a future challenge. There is no doubt that new experiments will expand our knowledge in terms of the relationship between DSB repair and chromatin architecture.

## Figures and Tables

**Figure 3 genes-13-00007-f003:**
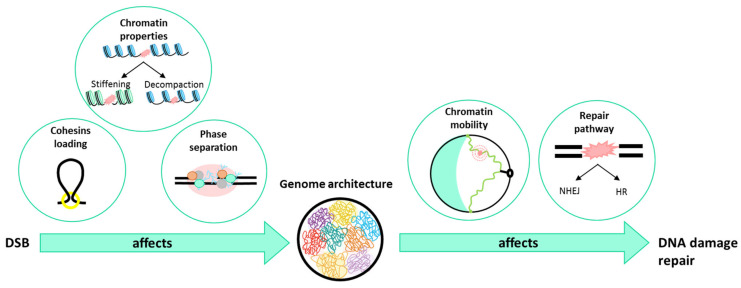
Relationship between DSB, genome architecture, and repair. After a DSB, different factors are involved. One of them is the loading of cohesins that confine repair factors by favoring *cis* contacts in mammals and yeast. A second factor concerns the changes in chromatin properties such as stiffening and/or decompaction. Finally, the recruitment of repair factors allows the formation of liquid droplets around the damaged locus. On the other hand, genome architecture affects DNA damage repair by increasing chromatin mobility upon damage differently, depending on the DSB position in yeast, and by influencing the choice of the repair pathway, depending on the damaged context.

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
