# Peer review of "3D Genome Organization: Causes and Consequences for DNA Damage and Repair"

_genes, 2021, doi:10.3390/genes13010007_

Round 1

Reviewer 1 Report

It is a well written genome organization review by Annia et.al. I have following comment for it.

1) Authors have not discussed causes for DNA damage and types of damages. they directly jumped to DSBs. There are so many other types of damages which alter chromatin and should be discussed here.

2) Modify the abstract. There are so many different types of damages other than DSB which occur and contribute to this effect. Authors specifically mentioned about DSB mobilization. Is there any particular reason for it? If it is then they should explain it.

3) DSB is not spelled completely at its first instance.

4) This review is mainly based on SMCs whereas review title is more general which is misleading. Author should modify the title of the topic.

5) Author should discuss the consequences of defective genome organization in term of disease association. They should discuss the future prospect, important questions in these field and interventions if available for it.

6)There are some error in English e.g. figure 3 title there should not be full-stop after architecture and capitalized next letter.

Author Response

We thank the reviewer for his/her constructive comments  and describe our responses below.

  • Authors have not discussed causes for DNA damage and types of damages. They directly jumped to DSBs. There are so many other types of damages which alter chromatin and should be discussed here.

The review we present here is interested in the links between genome organization and DNA damage, focusing on DSBs. The focus on DSBs is an editorial choice related to our expertise.

We have now modified the abstract (see 2) and specified the type of damages in the introduction.

  • Modify the abstract. There are so many different types of damages other than DSB which occur and contribute to this effect. Authors specifically mentioned about DSB mobilization. Is there any particular reason for it? If it is then they should explain it.

3) DSB is not spelled completely at its first instance.

We have added a sentence explaining that the organization of chromosomes is based on the SMC complexes, which have a preponderant role at the chromatin level and more globally on the folding of chromosomes and on anchoring at the periphery. We feel the notion of 3D organization of the genome in the title, justified and difficult to change.

5) Author should discuss the consequences of defective genome organization in term of disease association. They should discuss the future prospect, important questions in these field and interventions if available for it.

We have added a sentence in the discussion on the links between mutations of SMC complexes and human diseases. We have not developed the translational aspects, which go beyond our competences.

6) There are some error in English e.g. figure 3 title there should not be full-stop after architecture and capitalized next letter. This    is    now    corrected.

Reviewer 2 Report

In this review, authors nicely describe relationship between chromatin architecture and DNA damage/repair. The article provides relevant information, and would be useful for other researches in this field.

Below are suggestions, how a paper could be improved 

1) Lines 24-27. Authors could add nucleosomal level of chromatin organization.

2) Lines 34-36. This sounds confusing, in a way, that a reader could conclude, that the listed  pathways only generate controlled DSBs. But there are other ways, like constant work of TOP2. This needs to be refrased. 

3) If authors want to provide evidence of more direct involvement of SMC complexes in compartment organization, they could cite a work done with archaea chromatin https://pubmed.ncbi.nlm.nih.gov/31539494/

Author Response

We thank the reviewer for his or her comments and have made the requested clarifications as described below

REVIEW 2

  • Lines 24-27. Authors could add nucleosomal level of chromatin organization

We added this scale of chromatin organization as such :

« At the scale of the chromatin, in addition to nucleosomal landscape, chromatin-loops result from the extrusion activity of structural maintenance of chromosomes (SMCs) complexes (consisting of cohesins, condensins and Smc5/6) »

  • Lines 34-36. This sounds confusing, in a way, that a reader could conclude, that the listed  pathways only generate controlled DSBs. But there are other ways, like constant work of TOP2. This needs to be rephrased. 

We have reworded it so that controlled DSBs appear as a different path from other types of DSBs (changes are visible on the resubmitted review).

3) If authors want to provide evidence of more direct involvement of SMC complexes in compartment organization, they could cite a work done with archaea chromatin https://pubmed.ncbi.nlm.nih.gov/31539494/

We have added the following sentence and references lines 226-227

“This function of SMCs in shaping chromosomes is remarkably conserved and extends to all kingdoms of life, including bacteria and archaea”.

Marbouty, et al 2015 ; https://doi.org/10.1016/j.molcel.2015.07.020,

Lioy et al 2018 https://doi.org/10.1016/j.cell.2017.12.027

Takemata et al, 2019 https://doi.org/10.1016/j.cell.2019.08.036